# Diagnosis of Bacterial Pathogens in the Urine of Urinary-Tract-Infection Patients Using Surface-Enhanced Raman Spectroscopy

**DOI:** 10.3390/molecules23123374

**Published:** 2018-12-19

**Authors:** Ni Tien, Tzu-Hsien Lin, Zen-Chao Hung, Hsiu-Shen Lin, I-Kuan Wang, Hung-Chih Chen, Chiz-Tzung Chang

**Affiliations:** 1Department of Laboratory Medicine, China Medical University Hospital, No. 2 Yu-Der Rd, North district, Taichung 40447, Taiwan; t6719@mail.cmuh.org.tw (N.T.); t5677@mail.cmuh.org.tw (H.-S.L.); 2Department of Medical Laboratory Science and Biotechnology, China Medical University, No. 49, Hsueh-Shih Rd, North District, Taichung 40402, Taiwan; 3College of Medicine, China Medical University, Taiwan, No. 49, Hsueh-Shih Rd, North District, Taichung 40402, Taiwan; koichilin@hotmail.com (T.-H.L.); superman771104@gmail.com (Z.-C.H.); ikwang@mail.cmuh.org.tw (I.-K.W.); ma273737@gmail.com (C.-T.C.); 4Division of Nephrology, China Medical University Hospital, No. 2 Yu-Der Rd, North district, Taichung 40447, Taiwan; 5Division of Nephrology, Asia University Hospital, No. 222, Fuxin Road, Wufeng District, Taichung 41354, Taiwan

**Keywords:** urinary tract infection, Raman spectroscopy, SERS

## Abstract

(1) Background: surface-enhanced Raman spectroscopy (SERS) is a novel method for bacteria identification. However, reported applications of SERS in clinical diagnosis are limited. In this study, we used cylindrical SERS chips to detect urine pathogens in urinary tract infection (UTI) patients. (2) Methods: Urine samples were retrieved from 108 UTI patients. A 10 mL urine sample was sent to conventional bacterial culture as a reference. Another 10 mL urine sample was loaded on a SERS chip for bacteria identification and antibiotic susceptibility. We concentrated the urine specimen if the intensity of the Raman spectrum required enhancement. The resulting Raman spectrum was analyzed by a recognition software to compare with spectrum-form reference bacteria and was further confirmed by principal component analysis (PCA). (3) Results: There were 97 samples with single bacteria species identified by conventional urine culture and, among them, 93 can be successfully identified by using SERS without sample concentration. There were four samples that needed concentration for bacteria identification. Antibiotic susceptibility can also be found by SERS. There were seven mixed flora infections found by conventional culture, which can only be identified by the PCA method. (4) Conclusions: SERS can be used in the diagnosis of urinary tract infection with the aid of the recognition software and PCA.

## 1. Introduction

Urinary tract infections (UTIs) are a common infection that can affect the urethra, urinary bladder, ureter, or kidneys. The majority of UTIs are not serious, but some can lead to a potentially life-threatening complication such as sepsis. The treatment of UTIs is complicated by the increasing prevalence and spectrum of antimicrobial resistance [1,2,3]. In current practice, empirical antibiotics are used before a bacterial-culture result is available. The used antibiotic is adjusted according to the identified pathogen and the obtained antibiotic susceptibility test. However, a conventional bacterial culture takes at least 24 h to return results, and even longer to obtain the result of an antibiotic susceptibility test [4]. Therefore, a more efficient way for bacteria identification and antibiotic susceptibility is necessary for patient treatment. A quick and accurate pathogen identification and antibiotic use can shorten treatment course and avoid bacteremia formation.

The serology method, genotyping, matrix-assisted laser desorption-time of flight mass spectrometry (MALDI-TOF-MS), and nanochips have been exploited recently to shorten the time needed for pathogen identification [5,6]. However, when bacterial concentrations in the specimens are too low for pathogen detection, these methods are restricted. Some of the above-mentioned bacterial identification methods [5,6] may require a bacterial culture before bacteria identification. Few of these methods can directly verify antibiotic susceptibility.

Raman spectroscopy is a spectroscopic technique that uses observe-vibrational or -rotational modes to provide information on molecular vibrations and crystal structures [7]. Raman spectroscopy uses a laser light source to irradiate the sample and then generates Raman-scattered light [8]. The shifting energy gives information about vibrational modes in the system. The Raman spectrum (Raman scattering intensity) depends on the vibrational and rotational states of the molecules. Surface-enhanced Raman spectroscopy (SERS) is a surface-sensitive technique that enhances Raman scattering by molecule adsorbed on rough metal surface or nanostructure [9]. SERS increases the Raman scattering signal intensity by an enhancement factor of 10^10^ to 10^11^. SERS using nanoparticles as Raman substrates for molecule detection is an application of Nano Chip technology.

Recently, nanochip technology has applied successfully in antibiotic susceptibility. Liu et al. used SERS to detect isolated bacteria from blood culture, which was subsequently cultured in a tripticase soy broth (TSB) medium [10]. They were able to find the minimal inhibitory concentrations of antibiotics for these bacteria. The SERS method is also limited by its disturbance of sample impurity substances, such as protein or white blood cells (WBCs) that can compromise detection sensitivity. Human urine samples, similar to peritoneal dialysate from peritoneal dialysis patients, have the advantage of including much fewer proteins than blood, which can prevent the interference of detection from these impurities [11]. As concentrations of bacterial pathogens in biosamples may be lower than the detection limit, a larger amount of samples and repeated centrifuge are necessary to increase bacterial concentration. Urine also has the advantage of a sufficient sample amount to be exempt from the procedure to enrich bacterial amounts. For example, the bacterial culture test is more time-consuming than SERS chip detection.

In this study, we used a cylindrical SERS made up of silver nanoparticles coated on the tip of a 2 mm polymethylmethacrylate rod (AC). We used these SERS chips to detect pathogens from the urine samples of UTI patients.

## 2. Results

### 2.1. Establishment of Reference Raman Spectrum

There were 108 samples retrieved from UTI patients for the study. Conventional culture medium-based hospital bacterial culture results showed that there were 97 samples with a single bacterial species isolated, and seven samples reported with mixed flora. The other four samples with three kinds of isolated bacteria were deemed as contaminated samples (Table 1) resulting from inappropriate urine sampling. The isolated bacteria were loaded on cylindrical Raman chips, and the resulting spectrum served as a reference spectrum for comparison by RM.View software, recognition software for data-processing analysis. We used RM.View to compare the Raman spectrum from urine samples with the Raman spectrum of the reference bacteria from a conventional urine culture. Among the 101 isolated bacterial infections, four samples with Raman signals were too weak to be identified so they had to be retested after sample concentration was repeated (Figure 1, repeated concentrated method).

### 2.2. Raman Spectrum of Bacteria from Unprocessed and Centrifuged Urine Samples

We first compared the bacteria spectrum from unconcentrated urine samples with a bacteria spectrum from the concentrated ones. The Raman shift patterns were the same from the same batch of urine samples without being centrifuged (unprocessed method) or with centrifugation at 13,000 rpm for 5 min (centrifuged method). Signal intensity from the unprocessed supernatant tended to be weaker than signal intensity from concentrated samples (Figure 2). Most of the Raman spectrum signals from the unprocessed supernatant (700 rpm centrifuge for 10 min), though weak, were still detectable except the four samples from which the pathogens could not be identified by the centrifuged method.

### 2.3. Effect of Repeated Concentration on Raman Spectrum

There were four samples with very low signal intensity of the Raman spectrum despite urine samples treated with a 13,000 rpm centrifuge for 5 min. We extended the 13,000 rpm centrifugation to 10 min twice to further increase sample bacterial concentration and the resulting Raman spectrum signal intensity (Figure 1). The signal intensities of bacteria from samples that underwent repeated centrifugation were stronger than those of bacteria from samples that were only once centrifuged at 13,000 rpm for 5 min (Figure 3). As in most of the samples, bacteria in the supernatant can easily be identified; we loaded the unprocessed urine supernatant upon Raman chips for bacteria detection.

### 2.4. Raman Spectra of Antibiotic-Susceptible and Antibiotic-Resistant Bacteria

We then conducted a spectral analysis of antibiotic-susceptible bacteria and their antibiotic-resistant counterparts. The spectra were very similar in antibiotic-susceptible and antibiotic-resistant bacteria. Some bacterial signal, a 729 cm^−1^ peak, for example, could be seen in both antibiotic-susceptible *E. coli* and antibiotic-resistant *E. coli ESBL* (Figure 4A). Similarly, 727 cm^−1^ peaks could be seen in vancomycin-sensitive *Enterococcus faecalis* and its vancomycin-resistant counterpart (*VRE*) (Figure 4B). It is not easy to differentiate the antibiotic-susceptible from the antibiotic-resistant strain by only using the Raman shift spectrum.

### 2.5. Differentiation between Antibiotic-Susceptible and Antibiotic-Resistant Bacteria Strains Using Principal Component Analysis (PCA)

To identify the difference between antibiotic-susceptible and antibiotic-resistant strain, PCA was used [12]. We pooled all the digital signals of antibiotic-susceptible strains and their antibiotic-resistant counterparts. The antibiotic-susceptible and antibiotic-resistant bacteria formed two different groups of spots in the PCA plot. Therefore, PCA could be used to differentiate the signal of antibiotic-susceptible bacteria from that of an antibiotic-resistant strain. The PCA plots of *E. coli* versus *E. coli ESBL*, and *Enterococcus faecalis* versus *VRE* are demonstrated in Figure 5C,G.

### 2.6. Antibiotic Effect on Raman Spectra

To examine the drug-resistant bacteria detected by SERS chips, an antibiotic with a concentration higher than the minimal inhibitory concentration (MIC) was added to the urine samples [10]. The results of the Raman spectrum in the time-course study showed that the 729 cm^−1^ peak in *E. coli ESBL* disappeared after gentamicin treatment, but the 729 cm^−1^ peak could not be eliminated after the use of cefazolin (Figure 6A). Similarly, the use of vancomycin in the time course could contribute to the eradication of the 727 cm^−1^ signal peak in *Enterococcus faecalis*, but not to the annihilation of this peak (Figure 6B). The persistent presence of specific Raman signals after a therapeutic dose of antibiotic treatment indicated the presence of antibiotic-resistant bacteria.

### 2.7. Diagnosis of Mixed-Flora Infections

To solve the problem of mixed-flora infections, we used the PCA approach to perform the diagnosis. Among these mixed infections, infection causes could not be identified in seven cases. The Raman spectrum of mixed-flora infections resulted from a combination of the signal from different single flora. Some, but not all, bacteria-specific signals may be detectable. *Citrobacter*-specific signals (731 cm^−1^) and *Proteus*-specific signals (727 and 1133 cm^−1^) can be seen in samples with isolated bacterium infections, respectively (Figure 7B,C). However, not all bacterial signals could be identified in the Raman spectrum from specimens infected with *Proteus* and *Citrobacter*. It was difficult to identify a pathogen with the assistance of the software (Figure 7A). Therefore, we retrospectively used four kinds of known bacteria (*E. coli*, *Pseudomonas*, *Proteus*, and *Citrobacter*) and made a reference PCA plot. The four bacteria were located in different corners of the PCA plot. The location of patients infected with both *Proteus* and *Citrobacter* in the PCA plot was near the location of *Proteus* and *Citrobacter*, rather than near the location of *E. coli* or *Pseudomonas* (Figure 7F). The same tendency for the location of mixed infections can be seen in all seven cases (Appendix A).

## 3. Discussion

In this study, we used the Raman SERS technique to detect pathogens in the urine of UTI patients. We could identify bacterial pathogens in most urine samples after simple urine-sample centrifugation (i.e., 700 rpm for 10 min). In some cases, urine bacteria could be found only after repeated sample concentration.

Raman SERS is a culture-free method for pathogen identifications. SERS can enhance Raman scattering intensity 10^10^- to 10^11^-fold, which makes it possible to detect bacteria in samples without a predetection culture [13]. Therefore, the SERS technique has been used to detect bacterial infections in several studies [14,15,16,17] and has the advantage of quick, within-minutes diagnosis of the pathogen. Fast and correct pathogen diagnosis can shorten the treatment time, avoid unnecessary patient complications, and reduce treatment costs [18]. Urine is an aqueous sample, which frequently leads to the indeterminate spreading of samples loaded on a plate-shaped SERS chip, and causes poor reproducibility of study results [19]. Water in the fluid has to be evaporated to allow the contact of bacteria with the Raman substance coated on the chips [19]. In this study, we used cylindrical SERS chips that can easily make spontaneous contact of urine bacteria with the SERS substrate, and enhance detection sensitivity and reproducibility [6].

By comparing the Raman spectrum of standard bacteria with the Raman spectrum obtained from patient urine, we can predict the urine pathogen. With the assistance of the recognition software, pathogen identification becomes much easier and time-saving [11]. The amplitude of the Raman signal peak can be affected by bacterial concentration [20]. We cannot discriminate between different bacteria by SERS intensities as was shown in Figure 4; *E. coli* and *E. coli ESBL* cannot be differentiated simply by the signal intensities of similar peaks.

Methicillin-resistant *Staphylococcus aureus*, vancomycin-resistant *Enterococci* (*VRE*), and extended-spectrum beta-lactamase (ESBL)-producing bacterial are the common multiple-drug-resistant bacteria seen in UTIs [21]. Multiple drug resistance (MDR) increases the mortality and morbidity of UTIs [22,23], and also makes killing bacteria more difficult and challenging than before. A quick diagnosis of antibiotic susceptibility is therefore crucial for UTI treatments. SERS can reveal antibiotic susceptibility, though not as quickly as bacterial identifications. It is still much faster than the traditional disc-diffusion method. The gradual or time-dependent disappearance of a specific signal peak in the Raman spectrum of bacteria after antibiotic treatment indicates antibiotic susceptibility. The persistent existence of a specific signal peak after antibiotic treatments indicates an antibiotic-resistant pathogen. These two phenomena can be seen in Figure 6.

PCA analysis can also help differentiate an antibiotic-susceptible bacterial strain from antibiotic-resistant bacteria. *E. coli* spots clustered in the upper part of the chart, but *E. coli ESBL* grouped in the lower part of the chart (Figure 5C); *Enterococcus faecalis* in the left part, but *VRE* in the right part of the chart (Figure 5G). There were three *E. coli* spots located in the upper-right corner of the chart, and these spots appeared to be outliers (Figure 5C). Signal intensities of these three outliers were lower than those of the grouped spots in the left part of the chart (data not shown) and the lower signal intensities deviated these three spots from the main *E. coli* cluster. The lower signal intensities of the three samples may be due to low bacterial concentration in urine. The other possible causes of low signal intensities may result from inappropriate sample processing, for example, cell debris from pyuria attached to the bacterial cell wall, which can lead to poor light emitting and a subsequent low Raman signal.

PCA can also be used to facilitate the diagnosis of mixed-flora infections. The spot of mixed infections with two kinds of bacteria tends to be located near the spots of the two bacteria identified by conventional bacterial culture rather than near the other two unrelated bacteria. This tendency was also seen in all seven cases in which two kinds of bacteria were identified. The reason for this tendency is not clear.

The SERS technique, however, has some limitations. First, it may need an expensive apparatus to perform the experiment, such as confocal microscopy. We used a cylindrical SERS chip illuminated by a portable Raman spectrometer, which was relatively lower in price compared with that of the expansive confocal microscope used in other studies [6,11,14].

The second limitation of Raman SERS is that the concentration of bacteria in samples may be too low to be detected [20,24]. The limit of the lowest-detectable specimen bacteria concentration range was from 10^3^ to 10^5^ CFU/mL [20,25]. In this study, we chose bacteriuria urine samples from febrile patients with a UTI as our experiment model. Most of these patients were admitted via emergent department without a prehospital antibiotics treatment, and a routine hospital urine examination revealed the presence of bacteria by light microscope. The presence of pyuria and bacteria in routine urinalysis results suggested the presence of bacteria in the urine samples. Therefore, bacteria can be found in most samples without repeated centrifugation for sample concentration. Four urine samples without enough bacterial concentration for SERS detection and bacterial concentration can be increased by the centrifugation method [26,27]. Removing cells or cell debris in urine by 700 rpm centrifugation for 10 min, followed by concentration, can increase bacterial concentration (centrifugation of 13,000 rpm for 5 min) and enhance Raman signal intensity. Repeated concentration (centrifugation of 13,000 rpm for 10 min, twice) can further increase urine-sample bacterial concentration and improve the Raman spectrum resolution.

The third problem is mixed-flora infections, which can be found in some UTIs. Mixed-flora infections make bacterial identification difficult. We failed to find all single-pathogen-specific signals in the Raman spectrum of the mixed-flora infections. A Raman spectrometer with higher resolution may help us find some pathogen-specific signals of low intensity. We also need to establish a broad mixed-flora spectrum library to facilitate the recognition of software-assisted pathogen identifications.

## 4. Materials and Methods

### 4.1. Samples

Urine samples were obtained from febrile patients admitted to hospital with a urinary tract infection. None of these patients received antibiotic treatment before urine sampling. Urine analysis in these samples showed the presence of white blood cells (pyuria) and bacteria. A 10 mL urine sample was collected into a sterile container after periurethral orifice disinfection, sent to a conventional bacterial culture (culture medium-based), and the disc-diffusion method was used for antibiotic susceptibility. Meanwhile, 10 mL more of the same batch of the urine sample was sent for bacterial identification and antibiotic-susceptibility tests by the SERS chip method.

### 4.2. Sample Processing for SERS Studies

A urine sample of 10 mL was centrifuged at 700 rpm for 10 min. The supernatant was aspirated to separate it from the precipitate containing white blood cells (WBCs), immune cells, and cell debris. The unprocessed supernatant was loaded on a cylindrical Raman SERS chip (Labguide Co., Ltd. Taipei, Taiwan) (Figure 1, unprocessed method). For those unprocessed supernatants with a poor Raman spectrum resolution, we concentrated the supernatant (5 mL) by mixing it with an equal amount of distilled water and centrifuged the mixture at 13,000 rpm for 5 min; the supernatant after the high-speed centrifugation was discarded. The concentrated precipitate (3 μL) was loaded upon SERS chip for bacterial detection (Figure 1, centrifuged method). If the Raman spectrum resolution was still too low to identify a bacterial pathogen after first centrifugation, a 10 mL urine sample was treated with 700 rpm for 10 min, followed by twice centrifuging the supernatant at 13,000 for 10 min to further concentrate the urine sample. The final precipitate (3 μL) was then loaded on a Raman chip (Figure 1, repeat concentrated method).

### 4.3. Conventional Bacterial Culture

A calibrated inoculating loop was used to spread the urine sample on BBL TSA 5% sheep blood and EMB agar (Nippon Becton Dickinson Diagnostic Systems, Sparks, MD, USA). The isolated bacteria were identified using MALDI-TOF with a short-pulse laser (Bruker Daltonics, Billerica, MA, USA) as previously described [18,28,29,30]. Urine samples with 3 or more kinds of isolated bacteria were deemed contaminated.

### 4.4. Antimicrobial Susceptibility Test after Urine Culture

A Phoenix Automated Microbiology System with NMIC/ID-2 antimicrobial susceptibility kits (BD) (Becton, Dickinson Diagnostic Systems, Sparks, MD, USA) were used for antibiotic susceptibility test as we previously described. The study was performed following the manufacturer‘s recommendations [31].

### 4.5. Raman Spectrum of Different Bacteria

After loading the sample, the SERS chip was then put on a Raman spectrometer and illuminated with a laser light with a wavelength of 785 nm, a power of 20 mv, and a 5 s integration time (QE pro, Ocean, Dunedin, FL, USA) (OceanView, Version 1.5.0). Previously isolated known bacteria such as *E. coli* and *E. coli ESBL* from a conventional culture of our patient samples were also loaded on the SERS chip, and the resulting spectrum served as a reference Raman spectrum. Similarly, Methicillin-sensitive *Staphylococcus aureus* (MSSA), methicillin-resistant *Staphylococcus aureus* (MRSA), vancomycin-sensitive *Enterococcus*, vancomycin-resistant *Enterococcus* (VRE), and several kinds of gram-positive bacteria isolated from patients and confirmed by MALDI-TOF by a hospital bacterial laboratory were also chosen to make up the reference spectrum. AccuRam recognition software (Version:1.00.66) was used to compare the Raman spectrum from the UTI samples with a reference spectrum to facilitate pathogen identification [11]. A ≥95% fingerprint similarity between the spectrum of the sample and the reference spectrum was deemed as the same bacteria as a reference.

### 4.6. PCA

The Raman spectra of standard bacteria (400–2000 cm^−1^) and the spectra of bacteria after RM.View software were analyzed with PCA using SPSS software version 22 (IBM SPSS, Chicago, IL, USA). Dots in PCA that were colocalized with the dots from known standard bacteria were viewed as the same bacteria [12].

### 4.7. Antibiotic-Susceptibility Test

Antibiotics chosen for the susceptibility test were based on the identified pathogens and the antibiotic concentrations followed suggested concentrations, which were higher than the minimal inhibitory concentration [31]. Vancomycin and oxacillin were used for *Staphylococcus aureus*. Vancomycin was used for *Enterococcus* infection. Cefazolin, Ceftriaxone, Ciprofloxacin, and Gentamicin were used for gram-negative bacteria. For *Pseudomonas aeruginosa* infection, Ceftazidime was also used. Bacteria samples were mixed with antibiotics with different incubation periods. After incubation of a different duration, bacteria samples were loaded on SERS chips and the Raman spectrum was compared to the spectrum from the same batch of bacteria sample without antibiotic treatment.

## 5. Conclusions

SERS can be used to detect UTI pathogens without a laborious sample process. With the use of a cylindrical SERS chip and recognition software, bacteria in urine can be quickly identified. Antibiotic susceptibility can thereafter be obtained by the evolution of a bacteria-specific Raman signal peak. PCA can help differentiate antibiotic-susceptible bacteria from antibiotic-resistant bacteria. PCA may also be used to diagnose mixed-flora infections. Until now, SERS still cannot be used in clinical practice. There are not enough standard-reference Raman spectra from known bacteria for recognition software to 100% recognize patients’ urine samples. A comprehensive Raman library has to be established and more PCA plots from combinations of two different bacteria also need to be founded before SERS chips can be widely used in the clinical diagnosis of UTIs.

## Figures and Tables

**Figure 1 molecules-23-03374-f001:**
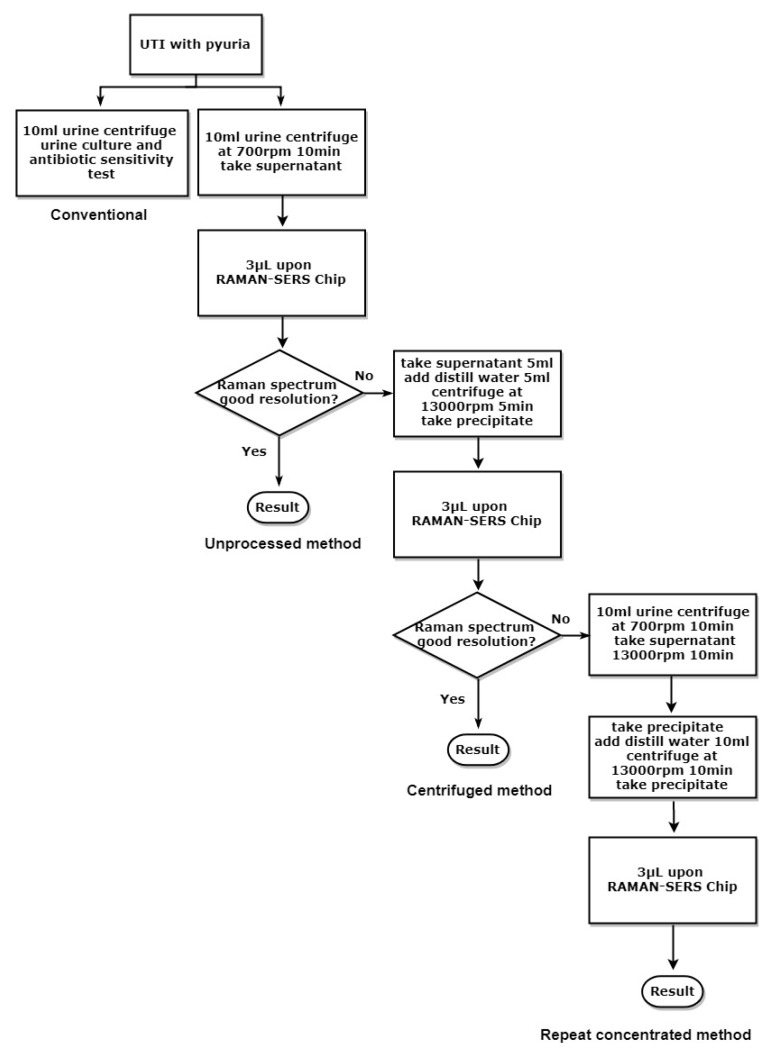
Flowchart of urine-sample processing.

**Figure 2 molecules-23-03374-f002:**
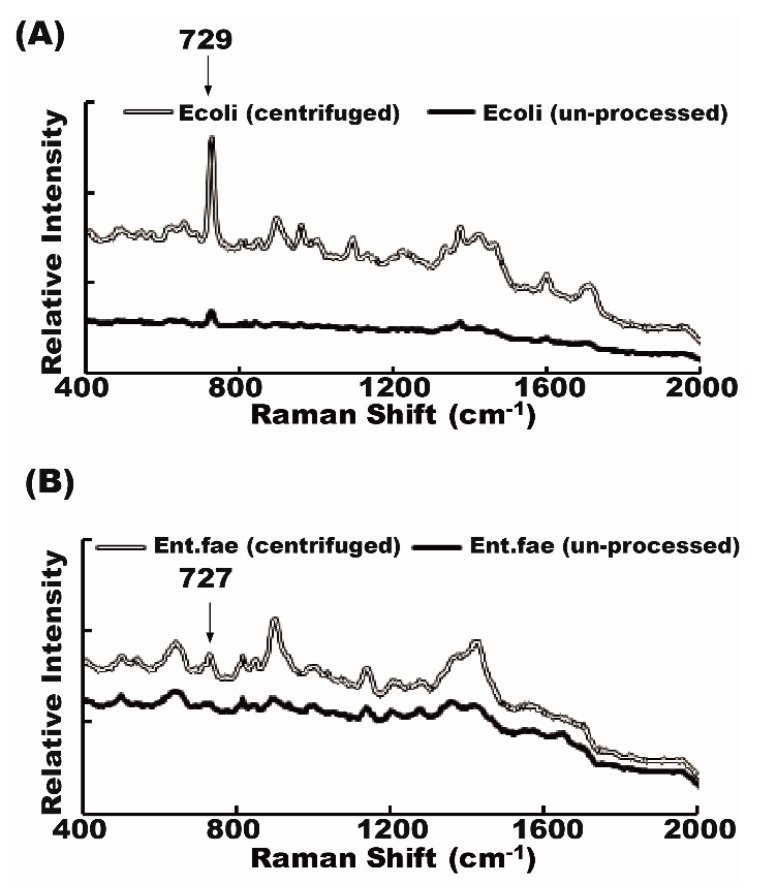
Raman shift patterns of bacteria from the unprocessed and centrifuged Raman. Raman spectrum of urine bacteria from the unprocessed sample (black line) and centrifuged sample (empty line) are similar. (**A**) *Escherichia coli*; (**B**) *Enterococcus faecalis*.

**Figure 3 molecules-23-03374-f003:**
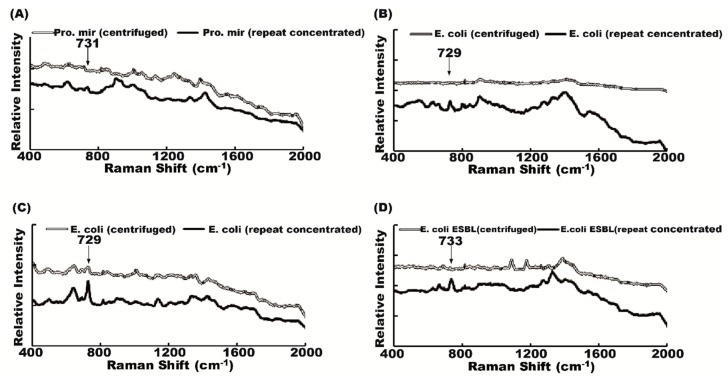
Centrifuged method and repeat concentrated method. Urinary-tract-infection (UTI) pathogens in the four samples that failed to be identified by the centrifugation method because of low resolution could be recognized with the repeat concentrated method. (**A**) *Proteus mirabilis*; (**B**,**C**) *E. coli*; and (**D**) *E. coli ESBL*.

**Figure 4 molecules-23-03374-f004:**
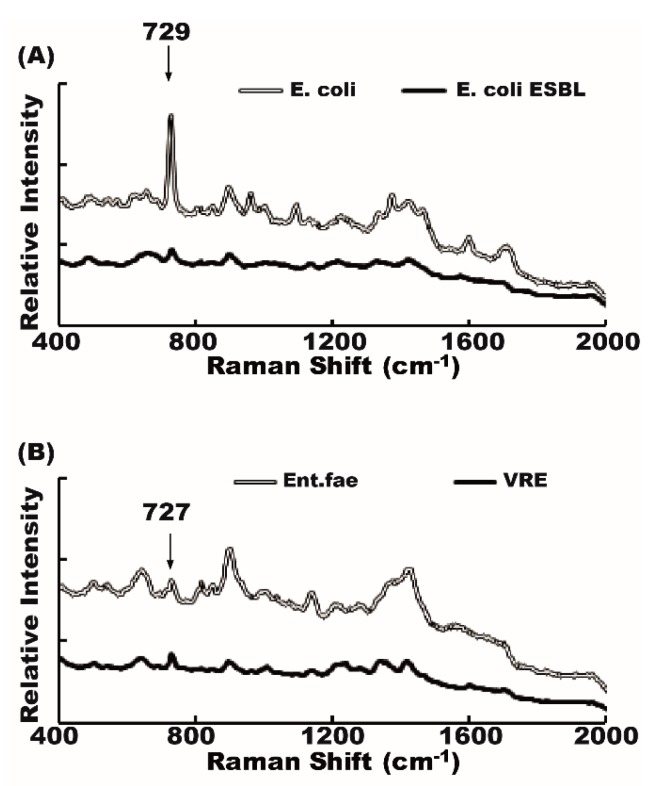
Raman spectra of antibiotic-susceptible and antibiotic-resistant bacteria. The Raman shift spectra of antibiotic-susceptible and antibiotic-resistant strains were similar. (**A**) *E. coli* and *E. coli ESBL*, (**B**) *Enterococcus faecalis* and vancomycin-resistant *Enterococcus* (*VRE*).

**Figure 5 molecules-23-03374-f005:**
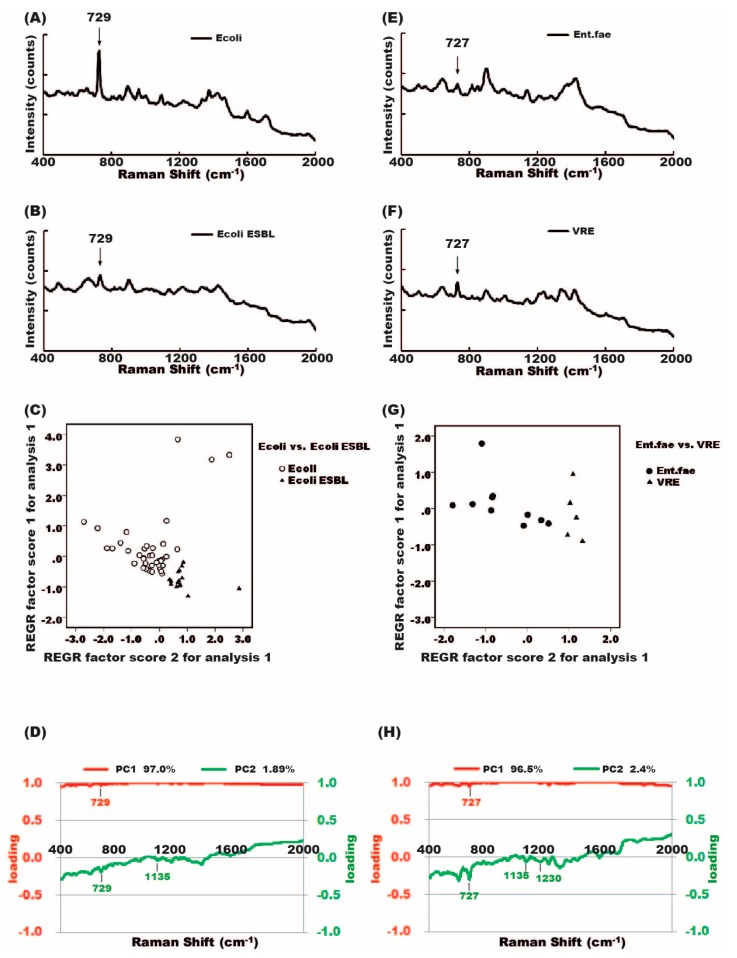
Principal component analysis (PCA) and the differentiation of antibiotic-susceptible and antibiotic-resistant bacteria. (**A**) Raman spectrum of *E. coli*; (**B**) Raman spectrum of *E. coli ESBL*; (**C**) PCA plots showed clustering of *E. coli* in the upper-left portion of the plot, and *E. coli ESBL* in the lower- right corner of the plot; (**D**) PC1 and PC2 loading plots corresponding to the PCA of (**C**); (**E**) Raman spectrum of *E. faecalis*; (**F**) Raman spectrum of *VRE*; (**G**) PCA plots show *E. faecalis* in the left side and *VRE* in the right side of the plot; (**H**) PC1 and PC2 loading plots corresponding to the PCA analysis of (**G**).

**Figure 6 molecules-23-03374-f006:**
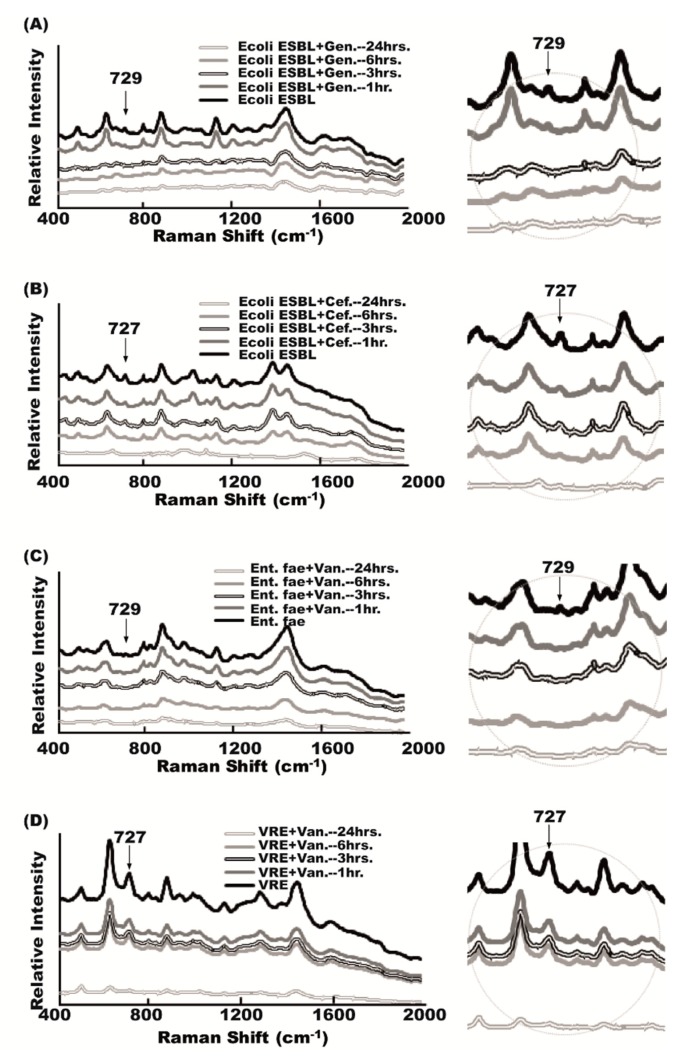
Antibiotic effects on bacterial-specific Raman signal. (**A**) 729 cm^−1^ signal peak of Raman spectrum time-dependently disappeared after gentamicin (Gen.) treatment in gentamicin-susceptible *E. coli ESBL* (gentamicin concentration: 0.256 mg/L); (**B**) *E. coli*-specific 729 cm^−1^ signal persisted in cefazolin (Cef.)-resistant *E. coli ESBL*. (cefazolin concentration: 0.256 mg/L); (**C**) *Enterococcus*-specific 727 cm^−1^ signal gradually disappeared after vancomycin treatment in vancomycin-susceptible *Enterococcus faecalis* (vancomycin concentration: 32.1 mg/L); (**D**) *Enterococcus*-specific 727 cm^−1^ signal persisted in *VRE* (vancomycin concentration: 0.256 mg/L).

**Figure 7 molecules-23-03374-f007:**
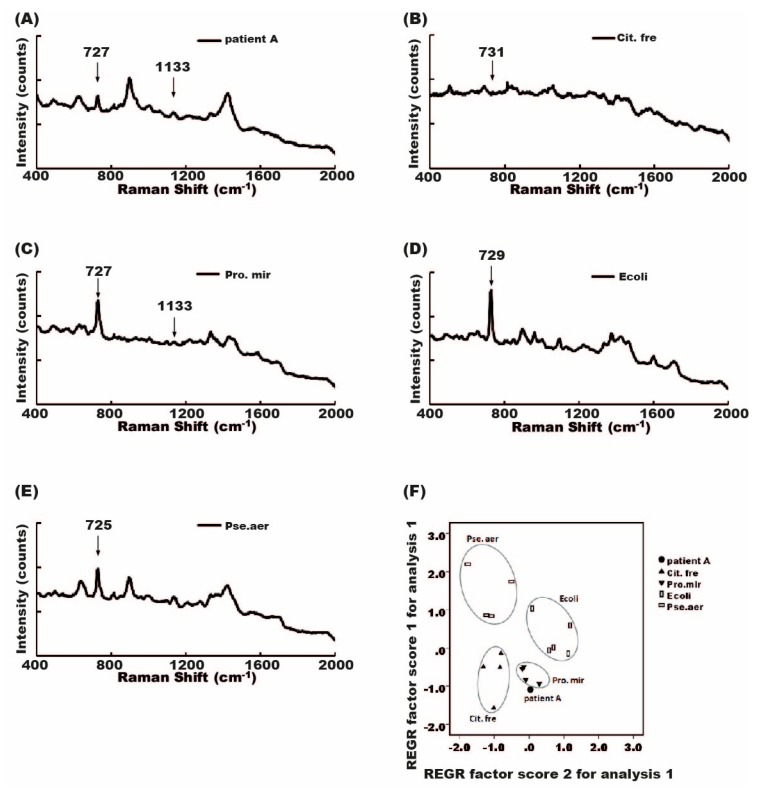
Diagnosis of mixed flora infections. Urine infected with both *Citrobacter ferundii* and *Proteus mirabilis* was loaded on Raman chip. (**A**) Signal peaks of 727 and 1133 cm^−1^ can be seen in the Raman spectrum from the urine of a patient with mixed *Citrobacter* and *Proteus* infection; (**B**) Raman spectrum of known *Citrobacter* in urine sample showed a specific signal peak at 731 cm^−1^; (**C**) Raman spectrum of *Proteus* showed specific signal peaks at 727 and 1133 cm^−1^; (**D**) Raman spectrum of *E. coli* showed a signal peak at 729 cm^−1^; (**E**) Raman spectrum of *Pseudomonas aeruginosa* showed a signal peak at 725 cm^−1^; (**F**) PCA showed that four different known bacteria were spotted in different locations of the plot, and the PCA-spot patient with mixed *Citrobacter* and *Proteus* infection was deposited near the locations of *Citrobacter* and *Proteus*; (**G**) PC1 and PC2 loading plots corresponding to the PCA of (F).

**Table 1 molecules-23-03374-t001:** Results of conventional culture and Raman surface-enhanced Raman spectroscopy (SERS) from 108 urinary tract patients.

	Conventional	Raman SERS
**Pathogen Result**		
Single bacteria	97	97
Mixed flora (two kinds)	7	failure to detect
Mixed flora (three kinds)	4	failure to detect
**Raman Sample Method**		
unprocessed method		93
repeat concentrated method		4

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
