# Peer review of "Diagnosis of Bacterial Pathogens in the Urine of Urinary-Tract-Infection Patients Using Surface-Enhanced Raman Spectroscopy"

_molecules, 2018, doi:10.3390/molecules23123374_

Round 1
Reviewer 1 Report
The authors presented the use of Surface-Enhanced Raman spectroscopy as a clinical tool to diagnose urinary tract infection.
The work presented by the authors is interesting and has some aspects of originality. English quality is extremely poor which makes it very difficult to fully understand the content of the manuscript. English is to be improved and the manuscript must be checked thoroughly for spelling and grammar mistakes. Before the manuscript can be accepted for publication, the following points should be addressed.
Abstract
It is not clear how from the results obtained the authors could conclude that SERS could be used as a diagnostic tool in urinary tract infection. Results need to be better presented (at the moment it is a list of results and it is not clear how they compare to established diagnostic techniques)
Line 25: Explain what “without or with sample concentration” means
Introduction
Line 39: Explain what “with or without associated with fever” means
Line 52: Explain what “without adequate sample bacteria concentration” means
Line 53: Clarify the sentence “some methods may require …. before bacteria identification”. Clarify what these methods are
Line 55: Briefly describe what “Nano-chip technique” is
Line 57: Explain what “tripticase soy broth” is
Line 63: Explain the meaning of “to shun the necessity to enrich bacteria amount”
Lines 65-68: Sentence “Methicillin-resistant ….. resistant bacteria seen in UTI” does not seem to fit within the section
Lines 68-70: Sentence “Multiple drug resistance …… challenging than before” does not seem to fit within the section
Lines 72-74: Description of the SERS is unclear in the sense that it is not explained if the SERS was on a lab-on-a-chip or it was a standard SERS setup
Results
Section 2.1
Line 77: Explain what “conventional culture results” authors refer to
Line 79: Sentence “the other 4 samples ….. contaminated samples” is incomplete and unclear
Line 81: Sentence “were served as …… by recognition software” is unclear. Explain what software was used
Line 83: Explain what “Figure 1 Repeat …. method section” means. What does “concentrated Raman” mean?
Table 1: Table is on two pages
Section 2.2
“Un-processed” should be changed into “unprocessed”
Line 87: Explain what “with and without sample concentration” means
Line 90: Explain what “unprocessed supernatant” means
Section 2.3
“Un-discernible” should be changed into “undiscernible”
Every reference to a figure is followed by a sentence (e.g. Figure 1 Repeat concentrated Raman). Why? What does this mean?
Lines 95-97: Sentence “to solve …… Raman sample” is incomplete and unclear
Line 101: What does “if without specific mention” mean?
Section 2.8
Why are all figures gathered in one single section?
Figure 5A: E. Coli shows a large and wide spread cluster. Could the three points on the top-right corner be outsiders? Explain further
Figure 6: The Legend covers some of the Raman peaks
Figure 7: Caption is on next page. In Fig. 7D there are just a handful of points per sample, this does not seem a good representation of the trend of the samples. Some more points are needed.
Discussion
This section is confusing as it is not clear if the results obtained were good or not. Additionally, the results were not compared with already existing literature. A better and more through discussion of the data is needed.
Conclusions
No convincing conclusions are drawn. Moreover, the authors claim that their findings showed that SERS can be used as a diagnostic tool but this is not clearly confirmed by their discussion of the results.
Author Response
Dear reviewer:
Thank you very much for reviewing our manuscript. We have followed your comments and made point-to-point correction in the following question and answers. We have engaged the help of a native English speaker to edit the text as suggested. Our changes were put in red characters in the marked copy file.
Hung-Chih, Chen
Reviewer1
Abstract
Line 25: Explain what “without or with sample concentration” means
Response to the comment: 10 ml sample without sample concentration by centrifuge was loaded upon Raman SERS chip (without sample concentration). If the signal intensity of resulting Raman spectrum was too weak for bacterial identification, we concentrated sample by repeat centrifuged and loaded the concentrated sample upon SERS chip (with sample concentration). We rewrite the sentence. (line 25-26 p.1)
Introduction
Line 39: Explain what “with or without associated with fever” means
Response to the comment: The majority of UTIs are not serious, but some can lead to a potentially life-threatening complication such as sepsis. We rewrite our sentence (line 39-40 p.1)
Line 52: Explain what “without adequate sample bacteria concentration” means
Response to the comment: When the bacterial concentrations in specimens are too low for pathogen detection, these methods (MALDI-TOF-MS and Nano-chip method) were restricted. We rewrite this sentence (line 50-53, p.2).
Line 53: Clarify the sentence “some methods may require …. before bacteria identification”. Clarify what these methods are
Response to the comment: Bacterial culture may require for bacteria identification. We rewrite this sentence (line 53-54, p.2).
Line 55: Briefly describe what “Nano-chip technique” is
Response to the comment: We add the paragraph for describe Raman, SERS and Nano-Chip technique (line 54-62, p2).
Line 57: Explain what “tripticase soy broth” is
Response to the comment: Trypticase soy broth is the growth media for the conventional culturing of bacteria. We add the word “medium” for it. (line 65, p.2)
Line 63: Explain the meaning of “to shun the necessity to enrich bacteria amount”
Response to the comment: We rewrite this sentence (line 70-74, p.2).
Lines 65-68: Sentence “Methicillin-resistant ….. resistant bacteria seen in UTI” does not seem to fit within the section
Response to the comment: We delete this paragraph and move it to discussion section.
Lines 68-70: Sentence “Multiple drug resistance …… challenging than before” does not seem to fit within the section
Response to the comment: We delete this paragraph and move it to discussion section.
Lines 72-74: Description of the SERS is unclear in the sense that it is not explained if the SERS was on a lab-on-a-chip or it was a standard SERS setup
Response to the comment: We used lab-on-a-chip method, SERS chips for bacteria identification. We modify “SERS” to “SERS chips” (line 76, p.2).
Results
Section 2.1
Line 77: Explain what “conventional culture results” authors refer to
Response to the comment: It means conventional hospital culture (culture medium based) results. We rewrite the sentence in (line 80-81, p.2).
Line 79: Sentence “the other 4 samples ….. contaminated samples” is incomplete and unclear
Response to the comment: We rewrite this sentence in (line 83, p.2).
Line 81: Sentence “were served as …… by recognition software” is unclear. Explain what software was used
Response to the comment: RM.View software- a recognition software for data processing and analysis software used for Raman spectrum bacterial identification. We add this sentence in (line 85-87, p.2).
Line 83: Explain what “Figure 1 Repeat ….method section” means. What does “concentrated Raman” mean?
Response to the comment: We use two kinds of urine sample concentrating method. We named it as “Centrifuged Raman” and “Repeated concentrated Raman”. For avoiding these confusing words, we change the “centrifuged Raman” to “centrifuged method” and “repeat concentrated Raman” to “repeat concentrated method”. (line 89 p.2; figure 1, p.3; line 89, p.2 and line 98-104, p.3.)
Table 1: Table is on two pages
Response to the comment: We correct it as your suggestion.
Section 2.2
“Un-processed” should be changed into “unprocessed”
Response to the comment: We correct it as your suggestion.
Line 87: Explain what “with and without sample concentration” means
Response to the comment: We rewrite this sentence (line 98-101, p.2).
Line 90: Explain what “unprocessed supernatant” means
Response to the comment: Sample only treated with 700 rpm centrifuge for 10 mins. We add the explanation (line 103, p.4).
Section 2.3
“Un-discernible” should be changed into “undiscernible”
Response to the comment: We corrected it as your suggestion.
Every reference to a figure is followed by a sentence (e.g. Figure 1 Repeat concentrated Raman). Why? What does this mean?
Response to the comment: These words in reference will delete after reload our figures.
Lines 95-97: Sentence “to solve …… Raman sample” is incomplete and unclear
Response to the comment: We rewrite this sentence in (line 111-114, p.4).
Line 101: What does “if without specific mention” mean?
Response to the comment: We deleted these words for the confusing meanings. (line 116-118, p.4)
Section 2.8
Why are all figures gathered in one single section?
Response to the comment: We correct it and reinsert the figures by the journal guideline.
Figure 5A: E. Coli shows a large and wide spread cluster. Could the three points on the top-right corner be outsiders? Explain further
Response to the comment: Signal intensities of these 3 outliners were lower than the grouped spots. Low bacteria concentration and inappropriate sample processing may cause the result. We add the explanation in Discssion section (line 230-237, p.9).
Figure 6: The Legend covers some of the Raman peaks
Response to the comment: We correct the figure 6 (p.7 figure 6) as your suggestion.
Figure 7: Caption is on next page. In Fig. 7D there are just a handful of points per sample, this does not seem a good representation of the trend of the samples. Some more points are needed.
Response to the comment: We correct this figure and add more points for each sample. (p.8 figure 7)
Discussion
This section is confusing as it is not clear if the results obtained were good or not. Additionally, the results were not compared with already existing literature. A better and more through discussion of the data is needed.
Response to the comment: We rewrite the discussion section. (p.9-10)
Conclusions
No convincing conclusions are drawn. Moreover, the authors claim that their findings showed that SERS can be used as a diagnostic tool but this is not clearly confirmed by their discussion of the results.
Response to the comment: We rewrite the discussion and conclusion. (p9-11)
Reviewer 2 Report
In this manuscript, the secretions from bacteria which induce UTI were detected by SERS chip. In the same way as the antibiotic sensitivity test after the culture, the secretions from from a single bacteria was detected. Moreover, those from antibiotic-resistant bacteria or mixed flora were discriminated by using PCA. Therefore, the SERS and PCA analysis can be useful methods for the diagnosis. However, there are some problems as follows.
PCA was used for the discrimination between antibiotic-susceptible and -resistant bacteria. But, it seems that they can be discriminated by the difference in the SERS intensities as shown in Fig.4.
In Fig.5 and Fig.7D, not only the score plots, but also the loading plots must be added. It may be useful to compare the loading plots and the corresponding SERS spectra.
In Fig.6, the authors should enlarge the spectra around at 729 or 727 cm-1. To clarify the decrease of the SERS peak by the addition of antibiotic, the peaks should be normalized by the peak intensity without the antibiotic.
Since the all spectra are horizontally long, the SERS peaks are not prominent.
Author Response
Dear reviewer:
Thank you very much for reviewing our manuscript. We have followed your comments and made point-to-point correction in the following question and answers.
Hung-Chih, Chen
Reviewer 2:
PCA was used for the discrimination between antibiotic-susceptible and -resistant bacteria. But, it seems that they can be discriminated by the difference in the SERS intensities as shown in Fig.4.
Response to the comment: The amplitude of Raman signal peak can be affected by bacterial concentration. We cannot discriminate different bacteria (antibiotics-susceptible or antibiotics-resistant) by SERS intensities simply by signal intensities of similar peaks. We add sentence for it in discussion section (line 210-212 p.9).
In Fig.5 and Fig.7D, not only the score plots, but also the loading plots must be added. It may be useful to compare the loading plots and the corresponding SERS spectra.
Response to the comment: We correct the Figure 5 and Figure 7 as your suggestion. We add the Raman spectrum of each bacteria (Figure 5A, 5B, 5D, 5E) in Figure 5. We also add the Raman spectrum of E. coli. and P. aeruginosa (add Figure 7D and 7E) in Figure 7.
In Fig.6, the authors should enlarge the spectra around at 729 or 727 cm-1. To clarify the decrease of the SERS peak by the addition of antibiotic, the peaks should be normalized by the peak intensity without the antibiotic.
Response to the comment: We correct the Figure 6 and enlarge the specific peak of each bacteria over right side of the figure as your suggestion.
Round 2
Reviewer 1 Report
Thanks to the authors for taking on board the comments. They did try to improve the article. The English still needs checking and in section 2.3 the figure number is still followed by a sentence.
Author Response
Reviewer 1 comment:
Thanks to the authors for taking on board the comments. They did try to improve the article. The English still needs checking and in section 2.3 the figure number is still followed by a sentence.
Response to the comment: We recheck the English grammar and spelling as your suggestion. We modify the section 2.3 as your suggestion (line 118-120, p.4).
Reviewer 2 Report
This manuscript was revised. However, the authors must show the loading plots rather than the Raman spectra in Fig.5 and Fig.7D.
Author Response
Reviewer 2 comment:
This manuscript was revised. However, the authors must show the loading plots rather than the Raman spectra in Fig.5 and Fig.7D.
Response to the comment: We add the loading plots of PCA as your suggestion (Figure 5 in p.6 and Figure 7 in p.8).